# A Learning Environment to Promote Awareness of the Experiential Learning Processes with Reflective Writing Support

Chanakarn Kingkaew [1,2,*], Thanaruk Theeramunkong [2,3], Thepchai Supnithi [4], Pronsiri Chatpreecha [5], Kai Morita [1], Koji Tanaka [6] and Mitsuru Ikeda [1]

1   School of Knowledge Science, Japan Advanced Institute of Science and Technology, Nomi, Ishikawa 923-1211, Japan
2   School of Information, Computer and Communication Technology, Sirindhorn International Institute of Technology, Thammasat University, Pathum Thani 12000, Thailand
3   The Royal Society of Thailand, Bangkok 10300, Thailand
4   Artificial Intelligence Research Group, National Electronics and Computer Technology Center, Pathum Thani 12120, Thailand
5   Department of Digital and Information Technology, Panyapiwat Institute of Management, Nonthaburi 11120, Thailand
6   Department of Psychological Science, College of Informatics and Human Communication, Kanazawa Institute of Technology, Nonoichi, Ishikawa 921-8501, Japan
*   Correspondence: c-kingkaew@jaist.ac.jp; Tel.: +66-89-108-9811

**Abstract:** Promoting reflective thinking is becoming increasingly important in helping learners develop strategies to apply new information to unpredictable situations during their daily activities. Reflective writing, based on Kolb's experiential learning cycle, could be one method for promoting reflective thinking that allows learners to consider their experiences and transform them into transferable knowledge, which can be applied to new contexts. However, learners cannot sufficiently practice reflective writing; thus, they cannot learn from their experiences. Therefore, our primary goal is to support reflective thinking by providing writing support. This study presents a computerized learning-environment design that helps learners to master experiential learning concepts using reflective writing. The study demonstrates how the learning-support function enhances experiential learning and enables the desired learning process to be captured. This helps mentors to provide suitable support toward understanding experiential learning. We also demonstrate how the learning environment can help learners to master experiential learning. This design, which has the dual role of supporting and observing implicit thinking behavior, can then be applied to a meta-level thinking support framework in other problem domains.

**Keywords:** educational technology; experiential learning; knowledge science; learning behavioral pattern; learning environment; metacognition; reflection; reflective thinking; reflective writing; technology-enhanced learning

## 1. Introduction

Society is becoming increasingly complex as a result of rapid technological advancement. As information becomes more accessible and is constantly changing in response to current events, the need for critical thinking is increasing daily. Thus, it is becoming essential for educators to promote thinking skills during learning to help the learners develop strategies for applying new knowledge to the complex situations they encounter in their daily activities. One useful framework for promoting thinking skills is Kolb's experiential learning (EL) theory, which explains how experience can transform into understanding and knowledge [1]. EL allows learners to contemplate an experience or a situation and gain knowledge by promoting reflective thinking [2,3]. In EL, reflective writing goes beyond a simple summary or straightforward description of the facts [4,5]. It is complex and

purposeful, and it expresses reflective thinking [4,6,7], encourages learners to think [8], improves learning, helps with self-regulation, enhances the learning experience [9,10], and supports learners' professional development [11].

Previous research has focused on applying EL theory in various educational settings such as internships, classrooms, laboratory activities [12], vocational education programs, and distance learning programs to improve learning activities and outcomes [13–15]. Reflection requires cognitive discipline, which requires learners' active engagement with their experience and usually takes time and effort to perform appropriately [2,7].

Tanaka et al. [16] used EL-guided questions for paper-based reflective writing in EL to train new employees on how to think. In particular, they organized a hands-on, concrete experience (CE) and promoted reflective observation (RO) and abstract conceptualization (AC) to strengthen their understanding. Their study shows that reflective writing improves learning and can be used as a good communication tool between learners and mentors to support the EL cycle. In addition, our pilot study [17] developed an online, EL reflective writing system that allowed observations among students, mentors, and faculty allocated to different companies and extended the approach of using EL-guided questions as in [16] to help college students improve their business knowledge through an internship program. Both studies found that EL-guided questions for reflective writing can promote learning EL concepts. However, many learners face difficulty applying EL concepts in their daily experiences. Most learners cannot use EL concepts to structure their thinking because they lack the real-world experience of applying such concepts in practice. In other words, the learners cannot connect their CE with abstract EL concepts.

As EL theory ought to be considered a learning process rather than a learning outcome [1,18], the primary goal of learning EL should be to engage learners in a process that enhances their learning potential, including providing feedback on the effectiveness of their learning efforts. To help learners identify the gap between their current and desired performance levels in EL and pinpoint what needs to be changed in order to improve, mentors can provide feedback to advance learners throughout the EL cycle [16]. However, outcome-oriented feedback, which focuses on the endpoint of the EL process, is often only provided via reflective writing reports. By contrast, language provisions that allow learners to think deeper and become aware of EL concepts, thereby externalizing reflections in EL and enabling observations on how learners think, are essential for mentors to recognize the learners' status and support them throughout the EL process.

This study presents the design and development of a computerized learning environment that promotes metacognitive thinking by supporting reflective writing based on EL concepts, making learners aware of the EL processes. We designed learning support functions that offer learners the opportunity to learn how to connect their CEs with abstract concepts through reflective writing, which promotes metacognitive thinking through EL processes. The support function used to capture thinking behavior in the EL process is designed to help mentors diagnose the learners' learning status to better support those who need it.

The remainder of this study is organized as follows. Section 2 presents background information and related work. Section 3 presents the EL framework, providing an overview of this research and clarifying the goal of the learning environment and the role of the support functions and educational technology developed in this research. The design of the learning support functions is discussed in Section 4. Understanding the learner–mentor interaction and how the support function helps the learners learn EL is discussed in Section 5. Section 6 presents the conclusion and limitations and discusses future works.

## 2. Background

This section presents the three main theoretical frameworks used in this study. First, we present Kolb's EL theory, as it serves as a base theory for reflective writing. Second, we discuss metacognitive thinking—an essential component in EL—which serves as the goal of EL. Third, we present reflective writing, the primary method for teaching learners to

acquire EL concepts. Related research on applying EL and providing a reflective writing system to improve education are also discussed.

There are models we can use to instruct reflective-thinking learning. For example, Gibbs' reflective learning cycle [19] is a framework for structuring learning from experience. It is based on six stages: description, feelings, evaluation, analysis, conclusion, and an action plan. Rolfe, Freshwater, and Jasper's reflective model [20] is based on three straightforward questions (What? So what? Now what?), which guide learners in reflective writing to consider an event that has happened and its implications and consequences. Jasper's ERA cycle [21], which stands for experience, reflection, and action, is a simple framework based on building an understanding from experience, examining feelings, and deciding on next steps. While these models provide a valuable guide for reflection and focus on the cyclical nature of learning from experiences, each model takes different approaches. These include differences in the number of stages and how comprehensive each model is. We use Kolb's experiential learning model as a base theory because Kolb's theory involves the learner's internal cognitive processes, for which the acquisition of abstract concepts can be applied flexibly in different contexts [22]. This research aims to promote metacognitive thinking for the learner by connecting concrete experiences and abstract concepts, in which learners develop metacognitive thinking by themselves. By adopting Kolb's theory as the base model of this framework, this research aims to provide learners with the opportunity to develop and become aware of AC in EL so that they may apply and refine EL cycles autonomously at the meta level in daily learning situations.

### 2.1. Experiential Learning

Kolb's EL theory defines learning as "the process whereby knowledge is created through the transformation of experience. Knowledge results from the combination of grasping and transforming experience" [1] (pp. 50–51). EL involves knowledge building through four learning modes, represented by the four stages of the learning cycle, as is shown in Figure 1.

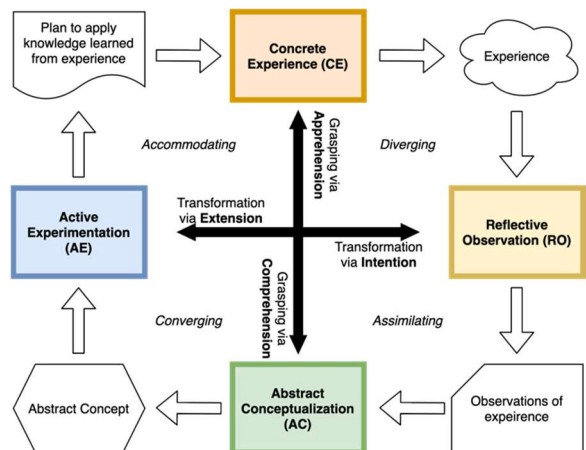

**Figure 1.** Kolb's EL cycle (adapted from [1,22]).

EL practitioners must have the ability to reflect on and observe their experience (RO), conceptualize observation as a logically sound theory (AC), use these theories to make decisions and solve problems (AE), and actively engage in a new experience (CE). A fundamental characteristic of EL is that knowledge is continuously derived from and tested through the learner's experience, or "what the learner learned in the way of knowledge and skill in one situation becomes an instrument of understanding and dealing effectively with the situations which follow" [18,22].

The EL cycle is a recursive progression through multiple learning cycles. This research defines *recursive thinking* as the transfer of previous learning experiences into new contexts.

This process is supported by a function that uses learners' past knowledge to promote actions that refine or revise that knowledge. By applying their past knowledge to new situations, learners can gain a deeper understanding of the concepts and skills they have learned. They can also develop the ability to apply this knowledge to a broader range of contexts. Thus, it is critical to support the process of transferring previous learning experiences into new contexts.

## 2.2. Reflective Writing and Metacognition

According to Flavell [23], metacognition refers to the understanding of one's own cognitive process. By encouraging learners to engage in metacognitive thinking during the four stages of Kolb's EL model, learners may become more aware of their learning process, which can result in the improved planning, monitoring, and evaluating of their learning process.

Tanaka et al. [24] found that the facilitation of discussion by teaching assistants in a short-term educational program that employs the EL model could promote metacognition; therefore, the EL cycle is considered to be a metacognitive activity.

Reflective writing is an effective method for promoting metacognitive thinking. Reflective writing support is widely accepted as an essential research issue for many subjects and cognitive skills. In principle, there are two primary focuses in using reflective writing; they are metacognitive-oriented and subject-oriented. For example, regarding the metacognitive-oriented focus, a study by Zarestky et al. [25] showed that reflective writing activity for multidisciplinary course design is a helpful tool to support students' metacognition and to foster self-regulated learning behaviors associated with a wide range of disciplinary topics and professional skills. Additionally, a study by Alt et al. [26] suggested that reflective journal writing can be adapted and integrated into different curricula as an instructional strategy for higher education, improving the quality of reflection in student journals and promoting lifelong learning skills.

By contrast, Franco et al. [27] found that fostering reflective writing in teaching communication skills encouraged experiential learning and group discussion among medical students.

In this study, we employ metacognitive support by fostering reflective writing with EL-guided questions to promote awareness of the EL process.

## 2.3. Reflective Writing as a Tool to Support Reflection

Numerous studies have investigated tools for supporting reflective writing for learners to document their learning and support their reflection. Dressler et al. [28] analyzed the writing style on Twitter, which provides up to 140 characters for reflective writing. The study found that writing concisely may be a practical method for reflecting on multiple experiences and emotions. Chanlin and Hung [29] encouraged students' reflective writing using an online internship journal system to facilitate review and retrospection. They found that most interns reacted positively to the learning and reflective processes embedded in the system. In addition, Moussa-Inaty [30] stated that providing guiding questions on reflective writing to learners before they write a reflection improves the quality of the reflection. Hence, ensuring that the learners use the EL cycle in their learning is essential, and following the EL process is vital for enhanced learning. As learners follow the EL cycle, they will develop EL-based thinking skills.

## 3. EL Framework

Before discussing the details of the support functions in the learning environment in Section 4, we outline the three main phases of the EL framework (see Figure 2):

- Pre-phase: understanding the EL concepts;
- Experiencing phase: understanding how to apply those EL concepts in reflective writing through a learning environment;
- Post-phase: reflecting on the self-monitoring of metacognitive thinking.

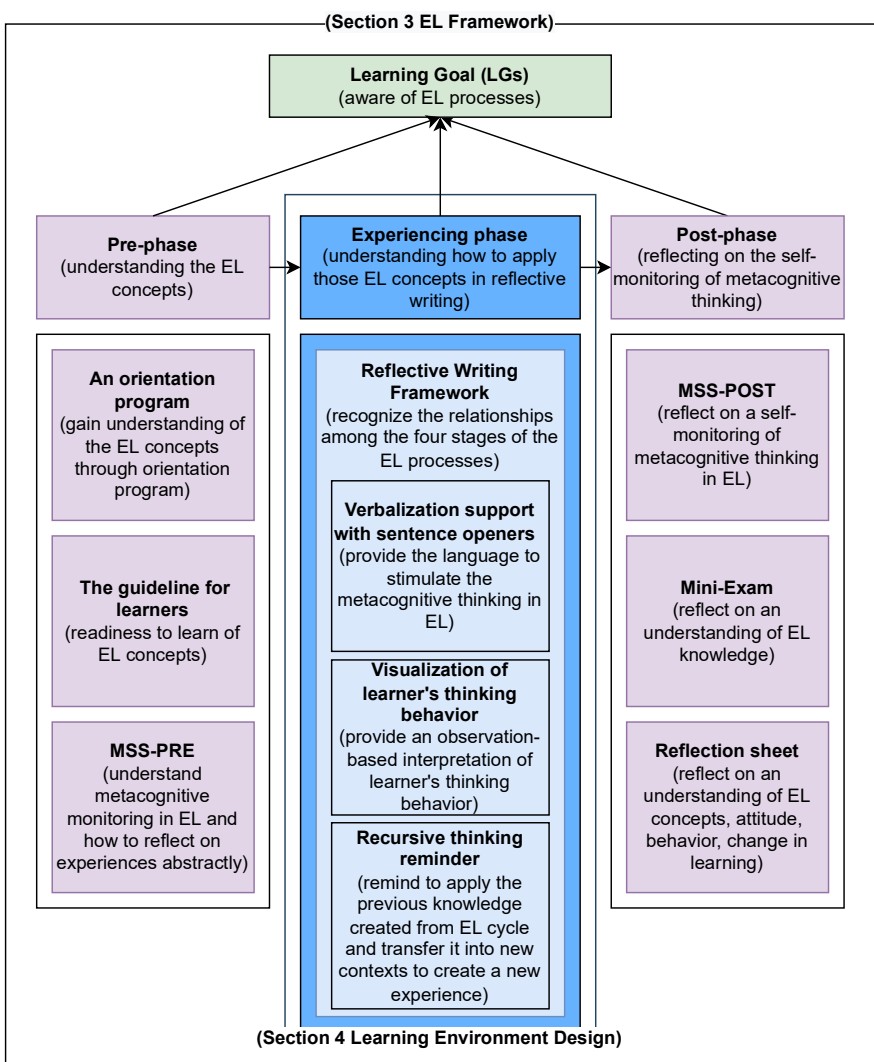

**Figure 2.** The relationship between the learning environment and EL framework.

*3.1. Pre-Phase*

In the pre-phase, we prepare related knowledge for learners to promote the readiness of EL. Learners understand what they are about to learn and use it in the experiencing phase. In this way, learners are motivated to think of their goals and learn about EL concepts. Thus, this phase aims to promote the understanding of EL concepts. The following educational programs can accomplish this goal in different ways.

3.1.1. An Orientation Program and Guideline for Learners

Before learning in the learning environment, learners must gain an understanding of the EL concepts through an orientation program. This orientation program is a short lecture that teaches learners about the thinking skills required to be active global citizens in the twenty-first century [31], enhancing their motivation to learn these skills. The program content is designed to promote the importance of supporting reflective thinking skills to understand the EL concepts and their necessity. This demonstrates how to perform reflective writing in the learning environment and explains the benefits of promoting metacognitive thinking. Moreover, the content of the orientation program is included in the guidelines for learners to prepare them for learning EL concepts, as EL can develop better skills if the learners have prior knowledge regarding their task [32]. When learners begin their reflective writing in the learning environment, they recall the lecture content upon reflection and realize the importance of knowledge by connecting it to experience.

3.1.2. The Metacognitive Skills Scale: BEFORE (MSS-PRE)

As was discussed in Section 2.2, we used the 30-question MSS for metacognitive thinking support, originally designed to assess university students' metacognitive skills [33]. However, in this study, we used the MSS as an educational tool to promote metacognitive thinking rather than as a scientific evaluation. This is because the MSS is dependent on the learners' reflection skills, and learners with poor meta-level reflection skills may overestimate their reflection ability. Additionally, this educational session is too short to see the skill development that is observed on the MSS scale.

By answering these 30 questions using the MSS-PRE, learners have a chance to reflect on their metacognitive skills, become familiar with self-monitoring concepts, and establish a goal to develop their EL skills.

Table 1 shows an example of how most learners may reflect on the MSS-PRE. As is shown in the first column, learners begin to reflect on their metacognitive skills in question Q.03, "I do not care about these questions. I just want to finish them first." Therefore, the MSS-PRE aims to improve learners' understanding of self-monitoring during metacognitive thinking.

**Table 1.** Example of an estimation of how learners might think during MSS-PRE and MSS-POST.

| Example Question in the MSS [33] | Learners' Reflections (MSS-PRE) | Learners' Reflections (MSS-POST) |
|---|---|---|
| I use my previous experience while organizing my new learning (Q.03) | I do not care about these questions. I just want to finish them first. But I realize that there is such a thing as using my previous experience to do something. | I am still unsure whether I did something like this in this class, but I guess I can do it well. |
| It is important for me to build meaningful relations between the learned subjects during learning (Q.33) | I am not sure about the meaning behind this item. I think I have never done this before. I will try to apply this skill in the future. | I am still unsure what the meaning of building meaningful relations between what I learn, and my experience is. Then, I guess I can do it. |
| I critically make a plan before beginning to study a text (Q.36) | I sometimes plan before studying a text; I think I should plan more often to improve my learning. | I think I always plan before studying a text but cannot remember it well. So, I guess it is ok to assess myself higher. |

*3.2. Experiencing Phase*

Given that practice is essential to developing metacognitive thinking [22], this phase aimed to enable learners to understand the application of EL concepts in their reflective writing. Learners practice reflective thinking through the computerized learning environment. In such an environment, they need to actively practice EL skills by connecting their real-world experience with EL concepts and obtaining support from mentors.

The reflective writing framework—the main learning support method in this study—provides the opportunity for learners to think by practicing reflective writing based on EL-guided questions. Under this framework, learners explicitly recognize the relationships among the four stages of the EL process (CE, RO, AC, and AE) through their reflective writing and can follow those stages to reflect on their experience. The reflective writing framework also comprises support functions to ensure that learners are able to learn EL concepts with adequate support. The following functions can accomplish this goal in different ways.

3.2.1. Verbalization Support with Sentence Openers

This function provides sentence opener options which, based on EL concepts, show the degree to which each sentence is similar. The various sentence openers allow learners to explore the connection with their experience by reflecting, self-questioning, reminding, making choices, and evaluating sentence opener options, allowing learners to think deeper. All these sentence openers can help suppress poor reflective writing through writing sup-

port. Therefore, this function provides sentence-opening *language* to reflect and stimulate metacognitive thinking in EL through verbalization.

### 3.2.2. Visualization of Learner's Thinking Behavior

This function is designed to indirectly support the learners by showing how they use sentence openers and visualizing it as a heatmap for mentors to demonstrate thinking tendencies. Mentors can interpret how learners change their thinking behavior from this visualization of writing style change. The learners are supported through feedback from mentors.

### 3.2.3. Recursive Thinking Reminder

To make learners recognize the importance of the recursive nature of an experience to make a linkage between EL cycles, this function aims to remind learners to apply previous knowledge created from the EL cycle and transfer it to new contexts to create a new experience, as was discussed in Section 2.1.

### 3.3. Post-Phase

This phase allows learners to reflect on their metacognitive thinking. Therefore, the purpose of the post-phase is not to evaluate learners but to act as an educational activity to promote reflection on EL.

### 3.3.1. The MSS:AFTER (MSS-POST)

While MSS-PRE prepares learners to learn metacognitive monitoring in EL and understand how to abstractly reflect on experiences, MSS-POST is used as an educational tool to promote reflection for learners to reflect on their monitoring of metacognitive thinking in EL. Table 1 shows an example of how learners might learn self-monitoring in EL. Learners acquire the EL concepts by practicing monitoring and controlling their learning activity through reflective writing in the learning environment. Subsequently, they reflect on their ability to apply the metacognitive skills in EL and improve their EL skills. One example is question Q.33, "It is important for me to build meaningful relations between the learned subjects during learning." In MSS-PRE, learners may not recognize the meaning of this skill, as is shown by the example of how they think: "I am not sure about the meaning behind this item. I think I have never done this before. I will try to apply this skill in the future." For MSS-POST, learners may reflect on the self-monitoring of metacognitive thinking in EL: "I am still unsure what the meaning of building meaningful relations between what I learned and my experience is. Then, I guess I can do it".

### 3.3.2. Mini-Exam

This mini-exam is introduced in the orientation program as a motivational tool to encourage the learners to learn the EL concepts. After the experiencing phase, the learners are asked to answer the following question: "How would you define EL in your words?"

### 3.3.3. Reflection Sheet

The reflection sheet consists of five-point graphic rating scales (very poor, poor, fair, well, and very well). This allows the learners to reflect on their understanding of the EL concepts (CE, RO, AC, AE, and recursive thinking). Furthermore, the reflection sheet includes open-ended questions that allow the learners to reflect on their attitude, behavior, understanding, and change in learning ability in this environment.

## 4. Learning Environment Design

In Section 3, we discussed the educational goals of our EL framework. In this section, we explain how each support function is designed to provide the learners with the experience of practicing EL through reflective writing to obtain those goals.

Research on computerized learning environments [34] typically aims to design learning objects representing real-world concepts (conceptual fidelity) and show the meaning of those concepts through simulation [35,36]. However, in metacognitive thinking training, no concrete object is shared with the learner, as only four text fields represent EL processes. In this learning environment, it remains unclear how to guide the learners and help them recognize the linkages among the four EL processes and the recursive processes between each EL cycle. Thus, the learning environment insists on learners' initiative and their active engagement in learning activities [37]. To clarify this point, we introduce functions that represent the connections among the EL processes, and we expect the learners to connect their experiences with the EL concepts to understand the meaning and motivate them to consider how they should think during each EL process. Here, we introduce the proposed computerized learning environment, which aims to support reflective thinking by providing writing support based on EL. Specifically, achieving the desired goal—as was discussed in the previous section—demands that two objectives be achieved: (i) a language for reflective writing is provided to promote reflective thinking based on EL concepts, and (ii) an observation-based interpretation of the learners' thinking behavior is provided.

As most learners find understanding the four EL processes challenging, our online reflective writing framework uses the idea of EL-guided questions [16,17] to enhance their understanding by following the list of EL questions. For this, we use text fields, each of which is represented by a question, as is shown in Figure 3:

- In the CE part, the question, "what have you experienced?" guides learners to search for their experience.
- In the RO part, the question, "what are your successful and failed experiences" allows learners to reflect on all their experiences, as many learners only focus on successful experiences and do not realize the shortcomings in their EL.
- In the AC part, the question, "what have you learned from the experience" guides learners to think of the abstract concept that represents what they have learned.
- In the AE part, the question, "what will you use for learning next" guides learners to consider how to apply what they have learned in the future.

In addition, in our user interface design, the output of one process becomes the input for the following process. This should allow learners to recognize the connections between the EL processes and the user interface. Figure 3 shows an example of the CE part, in which the learner applies a sentence opener (*italicized text*) in their reflective writing: "*I have significant experience of . . .* tried to develop a program based on object-oriented concepts of inheritance and understand this concept." This can be considered an output of CE, which then feeds into the RO part as an input (see the dotted line in Figure 3). In the RO part, "I succeeded at . . . able to understand the concept and write a program using the inheritance concept because . . . I carefully review my lesson and take time to conduct trial and error to write down the program".

Moreover, thinking support functions offer support for writing in three main ways, as was previously discussed: a verbalization support function with sentence openers, a visualization function of learners' thinking behaviors, and a recursive-thinking reminder function. The next subsections provide the design details of each support function.

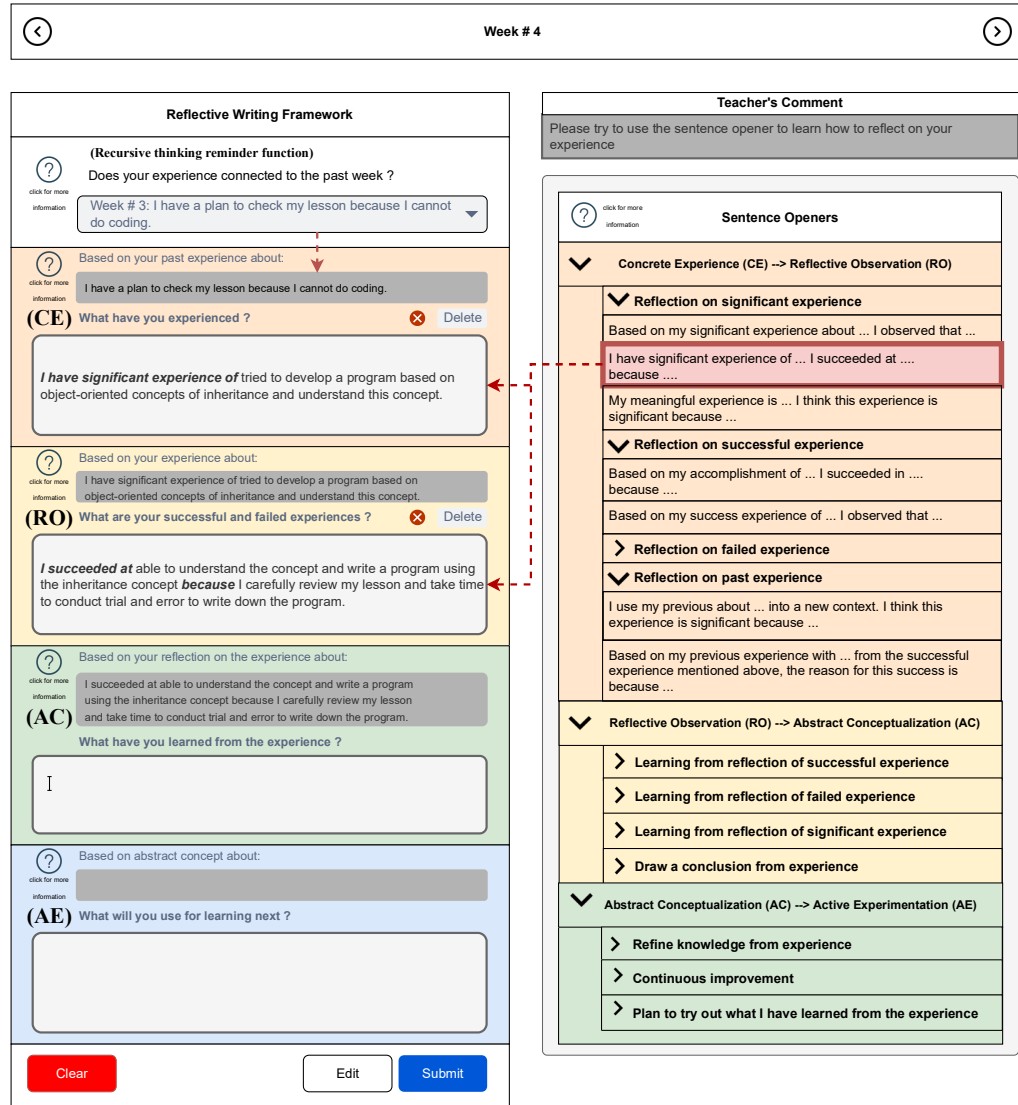

**Figure 3.** The EL environment interface for the learner.

### 4.1. Verbalization Support Function with Sentence Openers

As was discussed in Section 3, the verbalization support function with sentence openers provides language to stimulate metacognitive thinking in EL. Most learners have difficulty in understanding the four EL processes, abstracting the reflection of the experience, and self-monitoring the extent to which they use EL, as no language guides the learners to reflect on the EL concepts. Hence, by providing various sentence openers, learners can reflect on and evaluate their own thought patterns, which can help guide their thinking.

We adopt the following criteria for designing sentence openers. First, we collect the theoretical foundations for designing sentence openers. Second, we select suitable sentence-opener candidates based on the theory. Finally, we reorganize and categorize the sentences into groups.

In the first step, we surveyed the related literature to identify relevant sources for designing the sentence-opener candidates. The sources we used for this sentence-opener design were chosen to satisfy the requirement of promoting metacognitive thinking through reflective writing in EL.

- First, Kolb [1] provides the core EL concepts, and this serves as our primary source. This book describes EL-related concepts, such as gaining and transforming experience



and creating knowledge by connecting two processes (e.g., divergent, assimilating, convergent, and accommodating knowledge).

- Second, the MSS [33] has 30 questions on the metacognitive thinking required for EL. Hence, the MSS can guide the design of the sentence openers for the CE and AC concepts.
- Third, the concept of self-regulated learning by Zimmerman et al. [21,38] contains related knowledge on metacognitive thinking, such as forethought, self-reflection, self-control, and observational skills. Hence, self-regulated learning satisfies the requirement for sources as it is shown to reinforce the development of EL skills mutually [39].
- Fourth, Moon [5] describes what is not reflective writing and what kind of questions facilitate and prompt more profound reflection.

In the second step, we created a list of sentence openers based on the theoretical materials surveyed. We used the concepts extracted from the surveyed materials to consider what types of sentences represented those concepts (e.g., what kind of metacognitive thinking could be promoted by reflective writing). To promote the gain and transformation of experience, we applied this knowledge to guide the design of the sentence openers. Table 2 shows the examples.

- Concrete Experience (CE): The sentence opener promotes thinking by writing the concrete selection, thereby reminding the learners of a CE based on apprehension, which relies on the immediate and tangible qualities of the experience.
- Reflective Observation (RO): The sentence opener promotes thinking by writing the observation of CE based on intention, thereby reflecting internally on the various characteristics of their experiences and ideas.
- Abstract Conceptualization (AC): The sentence opener promotes thinking by writing the conceptual interpretation of the symbolic representation of experience based on comprehension, an individual's reliance on conceptual interpretation, and symbolic representation.
- Active Experimentation (AE): The sentence opener promotes thinking by writing the plan as a guide to creating new experience based on extension, thereby testing the ideas and experiences in the real world.

**Table 2.** Examples of sentence openers to promote thinking about EL concepts.

| Concept | Example of Sentence Opener | Source |
|---------|---------------------------|--------|
| Concrete Experience (CE) | <ul><li>The most significant experience was . . .</li><li>My meaningful experience is . . .</li><li>My successful experience is . . .</li><li>I use my previous experience about . . . into a new context (from [33] (Q.03))</li><li>I apply my knowledge about . . . to my new experience of . . .</li></ul> | (1) [1], (2) [33] |
| Reflective Observation (RO) | <ul><li>I am successful at . . . because . . .</li><li>The reason for my failure is . . .</li><li>I think this experience is significant because . . .</li><li>I succeed at . . . the reason is . . .</li><li>I observed that . . .</li><li>I failed to . . .</li></ul> | (1) [1], (2) [33], (3) [40], (4) [5] |
| Abstract Conceptualization (AC) | <ul><li>I can conclude that . . .</li><li>It is important to me to . . .</li><li>I learned from my experience that . . .</li><li>I can summarize that . . .</li><li>I crystallize that . . .</li><li>I develop a thought about . . .</li></ul> | (1) [1], (2) [33] |

**Table 2.** *Cont.*

| Concept | Example of Sentence Opener | Source |
| --- | --- | --- |
| Active Experimentation (AE) | • I will need to . . . to . . . <br> • I will . . . before I start studying. (From [33] (Q.36)) <br> • I will continue to . . . and apply this to . . . <br> • To fix my mistake about . . . I will . . . <br> • I will apply my success on . . . to . . . <br> • To continue to . . . I will need to . . . | (1) [1], <br> (2) [33], <br> (3) [40] |

To promote the knowledge gained from connecting the EL processes, the combination of grasping and transforming experience results in four fundamental forms of knowledge. We applied this knowledge as a guide on how to think about joining EL processes (Table 3 shows some examples of sentence openers):

- Concrete Experience (CE) to Reflective Observation (RO) (Divergent Thinking): grasps experience and transforms.
- Reflective Observation (RO) to Abstract Conceptualization (AC) (Assimilative Thinking): experience is absorbed through comprehension and transformed through intention.
- Abstract Conceptualization (AC) to Active Experimentation (AE) (Convergent Thinking): grasps via comprehension and transforms via extension.
- Active Experimentation (AE) to Concrete Experience (CE) (Accommodative Thinking): experience is grasped via apprehension and transformed via extension.

In the third step, we categorized the sentence openers into a hierarchy, as is shown in Table 3. The categorized sentences represent the structure of the user interfaces (Figure 3).

The various options for sentence openers require the learner to reflect, question, compare, and evaluate to choose the sentence opener that reflects their experience, encouraging them to think deeper. First, the learner chooses to use a particular option. For example, the phrasing "(CE) I have significant experience of . . . (RO) I succeeded at . . . . because . . . " is a combination of the CE to RO process with a reflection on the significant experience category. This sentence is split into CE and RO text fields (Figure 3). The learner must recall which experience to use as an input in CE and select the reflection of experience in RO using reflective writing. Subsequently, the learner will write what they think.

Therefore, one role of the sentence openers is as language to support writing for stimulating metacognitive thinking. Another role of the sentence openers is as learning sensors which enable observations of metacognitive thinking or implicit thinking phenomena in the learner's mind to observe how the learner thinks. We discuss this role in more detail in Section 4.2.

**Table 3.** Categorization of the sentence openers.

| Concept | Sub-Concept | Example of the Sentence Openers |
|---|---|---|
| CE to RO (Divergent Thinking) | Reflection on a significant experience | • (CE) The most significant experience was . . . (RO) I am successful at . . . because . . . <br> • (CE) I have significant experience of . . . (RO) I succeeded at . . . because . . . |
| | Reflection on a successful experience | • (CE) My successful experience is . . . (RO) with that experience I am successful at . . . because . . . |
| | Reflection on a failed experience | • (CE) Based on my failed experience about . . . (RO) the reason for my failure is . . . <br> • (CE) Based on my failed experience about . . . (RO) I failed to . . . |
| | Reflection on a past experience | • (CE) I use my previous experience about . . . into a new context. (RO) I think this experience is significant because . . . |
| RO to AC (Assimilative Thinking) | Learning from the reflection on a failed experience | • (RO) From my observation of my experience that . . . (AC) I can conclude that . . . |
| | Learning from the reflection on a successful experience | • (RO) The successful reason . . . (AC) I learned that . . . |
| | Learning from the reflection on a significant experience | • (RO) I think this experience is significant because . . . (AC) I crystallize that . . . |
| | Drawing a conclusion from an experience | • (RO) I am successful at . . . because . . . (AE) I can summarize that . . . |
| AC to AE (Convergent Thinking) | Refining knowledge from an experience | • (AC) From my conclusion that . . . (AE) I will need to . . . to . . . |
| | Continuous improvement | • (AC) I learned that . . . (AE) To continue to . . . I will need to . . . |
| | Planning to test what I have learned from the experience | • (AE) To fix my mistake about . . . (CE) I will . . . |
| AE to CE (Accommodative Thinking) | Recursive thinking | • (AE) I will need to . . . to . . . (CE) Based on previous learning about . . . <br> • (AE) To fix my mistake about . . . (CE), I will . . . <br> • (AE) I continue to . . . (CE), I will need to . . . |

### *4.2. Visualization of Learners' Thinking Behavior Function*

To observe learners' thinking behaviors in the EL process, we design a visualization of learners' thinking-behavior function for mentors. This visualization illustrates the learner's implicit thinking behaviors that occur in their mind. It is an indirect way to support the learner from a mentor by observing how they think and giving appropriate feedback based on the learner's status.

In one example in Woolf [41], the learning environment involves a cardiac-arrest tutor. In this environment, the learners' goal is to observe the vital signs and make good decisions based on their cardiac-arrest treatment knowledge. Learners can learn what they should change in detail (e.g., right or wrong procedures), as their knowledge clearly shows them how to perform operations. These failures allow learners to understand which sign is vital in which situation. The system can note their lack of judgment and provide them with feedback.

Meanwhile, for metacognitive thinking training in EL, learners aim to become aware of the EL processes. They try to learn to apply the EL concepts by connecting their CE to an abstract concept of EL. However, learners do not have an exact procedure for what they should do. Thus, it is challenging for them to clearly understand when and what EL knowledge to apply, as the learning process is implicit. Learners who desire an experience in CE determine whether this experience is good for abstraction. If they realize this, they retry the selection of CE. Therefore, trial and error are needed for them to understand the connection between the four EL processes and how to select a good CE for the EL cycle.

These procedures, such as how learners use sentence openers, can be a good clue for the mentor to support the learner. The learning sensor role of the sentence openers captures these. Such actions include:

- Clicking to expand the sentence opener at the root node at time t1;
- Browsing through the options for sentence openers at time t2;
- Writing text after the selection of the sentence opener at time t3.
- Changing sentence openers at time t4;
- Selecting the sentence-opener category (CE to RO) at time t5;
- Finally selecting the sentence-opener choice (# SO-2-3-1) at time t6;
- Deleting the sentence-opener choice (# SO-1-1-6) at time t7.

Each of these learner's event actions is represented by the four-tuples data structure (object, action, key, and time) or sensing data.

- Object: The learning object in the user interface with which learners interact, such as the sub-category of sentence opener in the CE and RO category, the CE's text field, the sentence openers list, and the recursive list.
- Action: An action value that is performed by learners. For example, they select or browse the sentence-opener options by typing, opening the list, and deleting.
- Key: The value that the learners input, such as the selected sentence opener, category of the sentence opener, choice of the previous week's experience, and use of text.
- Time: The period in which the learners' events happened.

The knowledge of how to recognize different types of learning behavioral patterns to determine types of behavior is essential for mentors to support learners. The sentence-opener learning behavior (SO) patterns can be categorized into five classes, shown in Figure 4a. We designed a coloring policy for their representation, and each EL concept is differentiated by color. The stronger cue may indicate the learner's confidence in applying metacognitive thinking.

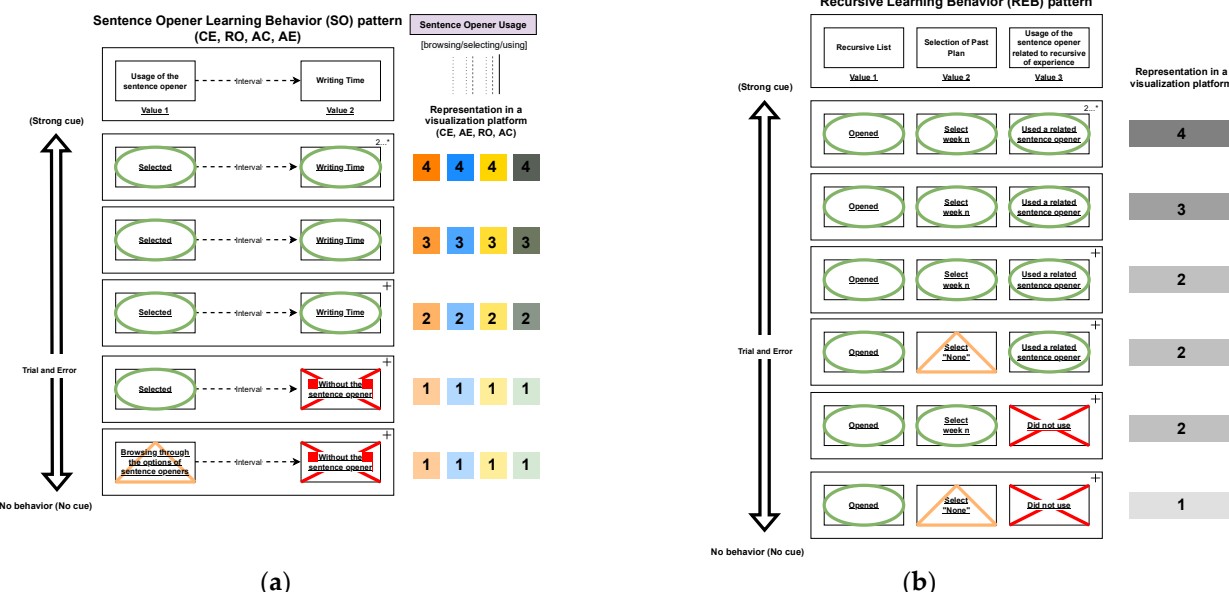

**Figure 4.** The prepared knowledge; representations of learning behavior patterns: (**a**) sentence opener learning behavior (SO) pattern (CE, RO, AC, AE); and (**b**) recursive learning behavior (REB) pattern.

- 4 (Strongest Cue) is when the learner selects a sentence opener for reflective writing for at least two consecutive weeks without trial and error.
- 3 (Strong Cue) is when the learner selects a sentence opener for reflective writing without trial and error for the first time.
- 2 (Neutral) is when the learner selects sentence openers for reflective writing with trial and error.
- 1 (Light Cue) is when the learner browses through sentence-opener options and does not use them in reflective writing.
- 0 (No Cue) is when the learner does not use sentence openers.

The Recursive Learning Behavior (REB) patterns can be categorized into five classes, shown in Figure 4b. A stronger cue may indicate the learner's confidence in using previous experience in new contexts. Please see the function details in Section 4.3.

- 4 (Strongest Cue) is when the learner selects a sentence opener in CE and chooses one of the previous experiences in a recursive list for at least two consecutive weeks without trial and error.
- 3 (Strong Cue) is when the learner selects a sentence opener in CE and chooses one of the previous experiences in a recursive list for the first time.
- 2 (Neutral) is when the learner chooses one of the previous experiences in a recursive list with trial and error.
- 1 (Light Cue) is when the learner chooses one of the previous experiences in a recursive list with trial and error.
- 0 (No Cue) is when the learner does not use a recursive list.

The sequence of learning events (object, action, key, and time), shown in Figure 5A, becomes the input into the function, These learning events, as sensing data, are then parsed into a pattern-matching process where matched data are filtered into a given class, as is shown in Figure 5B. The data are then visualized at a given time when the behaviors start and stop. The output of the function is the visual representation of learners' thinking behaviors in their minds, as is shown in Figure 5C. This represents how each learner learned to think about acquiring the EL concepts and how this learner spends time on each activity each week. Figure 5D shows an example of one interpretation that might come from a mentor.

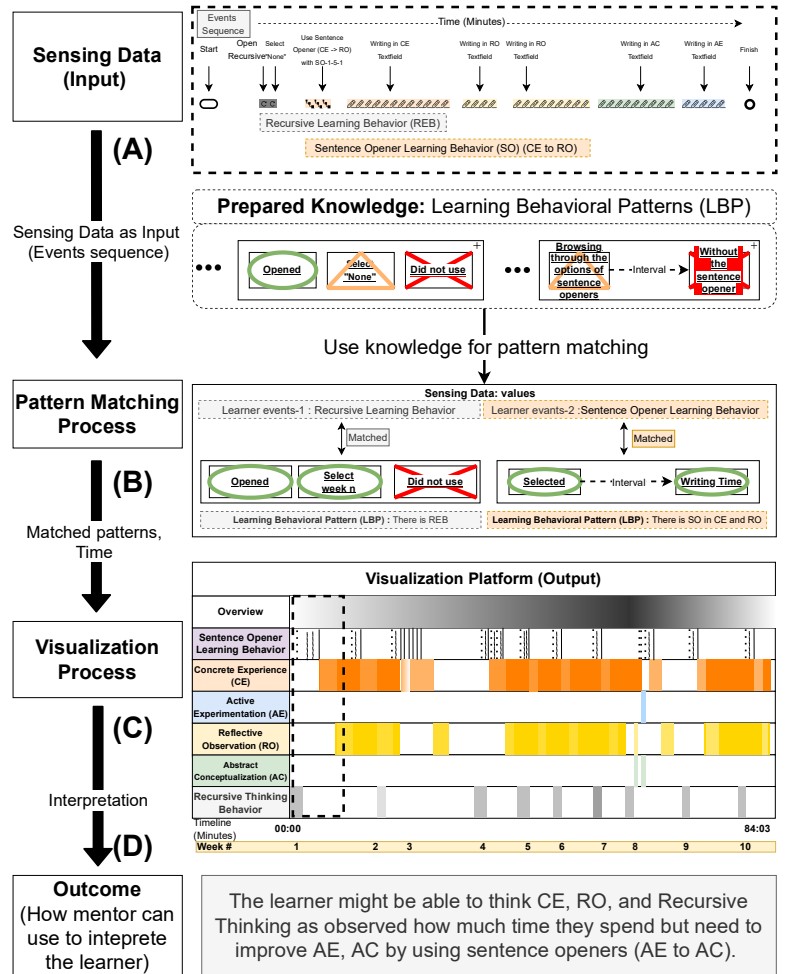

**Figure 5.** The visualization of the learner's thinking-behavior function process: (A) Sensing Data as Input; (B) Pattern Matching Process; (C) Visualization Process and (D) Outcome as mentor's interpretation.

As language, the sentence openers play a dual role in thinking support and observation.

### 4.3. Recursive-Thinking Reminder Function

The recursive-thinking reminder function is a user interface comprising a drop-down list that takes the learning experiences of the EL cycle, written in AE from previous weeks and displaying past plans, the knowledge learned, and the goals set. Learners set goals to test existing ideas in AE and thus make modifications, which can be fed as a new experience into the future steps due to the recursive nature of learning.

Learners' past experiences are listed to remind them of what they have previously learned and encourage them to apply it to new contexts. This list is placed at the top of the reflective writing framework containing the previous week's planned activities to help learners understand how to connect past experiences to the present, as is shown in Figure 3. Therefore, recursive learning behavior starts when learners use this function.

As is shown by the example in Figure 6, a learner can open the list of previous experiences written in the past week. In Week four, the list shows their choices of previous experience in the past three weeks. The learner questions and searches for an experience in which they were engaged in Week Three: "I have a plan to check my lesson because I cannot do coding." The choice "New Experience" shows that the learner may not yet have a matching previous experience, meaning they learned something from a new experience.

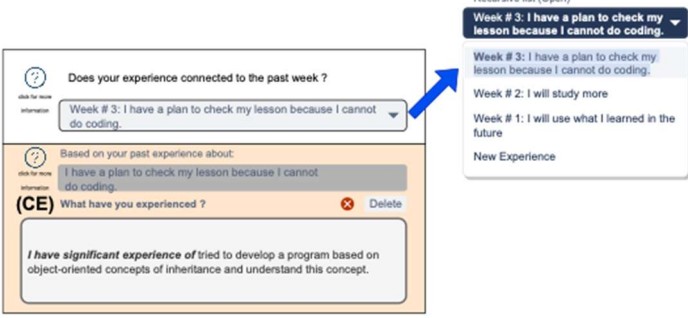

**Figure 6.** The recursive-thinking reminder function.

## 5. Exemplifying EL in the Learning Environment

In this section, we show how learners learn the EL concepts in this learning environment to promote metacognitive thinking in EL through reflective writing. Mentors use the learners' data generated in this learning environment to provide them with appropriate feedback. Figure 7 shows the experimental design, which distinguishes the data needed to learn in this learning environment. In addition, we survey learners' and mentors' perspectives to understand their impressions of this learning environment.

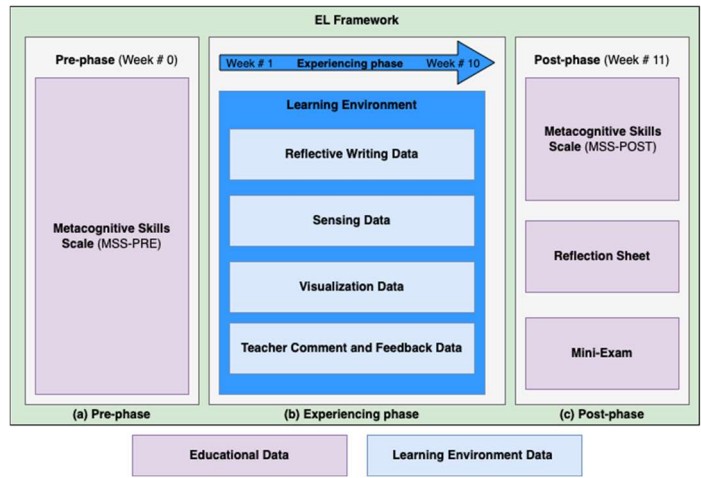

**Figure 7.** Experimental design with the educational data and learning environment data.

In this study, the participants were 70 students (hereafter "learners") from Panyapiwat Institute of Management. This institute has a novel curriculum in which the learners have a long internship (lasting three to nine months) in multiple phases that spans from their first year to their fourth year. Learners were enrolled in an object-oriented programming course in a distance-learning setup. We conducted the research with this sample because of the nature of the course, which requires metacognitive thinking [42]. Learners were taught how to analyze problems, represent the solution using different strategies, and verify the solution. Next, they needed to convert the problem solution into a program using a programming language and then evaluate their program for syntactical and logical errors to ensure that the output solved the problem. This required learners to go through repeated trial-and-error attempts, which continued until they successfully solved the programming problems. These activities taught them the thinking behavior needed to transfer their acquired problem-solving skills to new contexts. As they will need to apply these skills during their multiple-phased internships, EL skills are essential for them to integrate the knowledge gained in the classroom with the practical experience found in the workplace.

The orientation program introduced all learners to the learning environment, the guidelines were given, and the learners were asked to complete the MSS-PRE. Learners

were informed that there would be a mini-exam, reflection sheet, and the completion of the MSS-POST after finishing the class. The learning was conducted via distance learning due to COVID-19 restrictions. The study period lasted ten weeks; at the end of each week, all learners were required to complete reflective writings about their learning experiences. Each learner was given an account they could access through a web-based online learning environment. These writings and the mini-exam counted toward ten percent of their final score to encourage learners to carry out the reflective writing component of the course. The teacher of the object-oriented programming subject also served as a mentor to support EL. The mentor was asked to identify the learners' problems and provide comments and feedback to learners each week.

We selected two learners for our case studies, which aim to show the learner–mentor interaction, demonstrate our designed learning environment, show how the proposed support function helps learners to grasp EL, and describe how the mentors used the data generated from the learning environment to support learners. On the one hand, one learner may be fully aware of the EL concepts (CE, RO, AC, and AE) with the support of the functions designed in the learning environment. On the other hand, as in another case, the mentor could observe that the learner may be partially aware of the EL concepts (CE and RO) and detect an overestimation of their knowledge. In the next section, we show case study #1, in which the learner became fully aware of the EL processes, followed by case study #2, in which the learner was partially aware of the EL processes.

### 5.1. Case Study #1: The Learner That Became Aware of EL Processes (CE, RO, AC, and AE)

In case study #1, this learner gradually became aware of the EL processes (CE, RO, AC, and AE) with support from the learning environment and their mentor. The visualization in Figure 8 shows the visualization of the learner's thinking behavior from visualization platform.

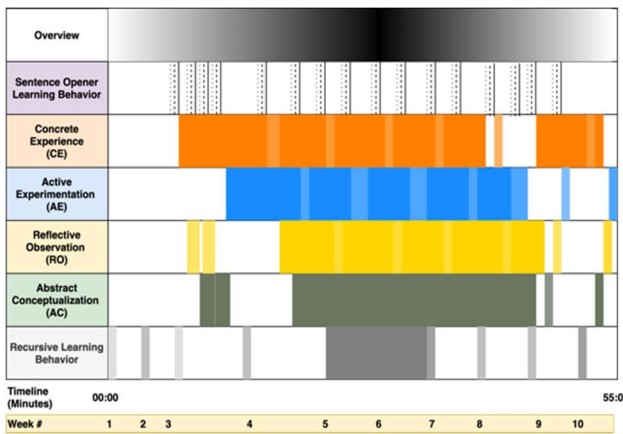

**Figure 8.** The visualization platform shows the learner's thinking behavior in case study # 1.

### 5.1.1. Interpretation of Learners in Case Study # 1

From the data observation, the mentor observed that the learner wrote their reflective writing during the first week without using the sentence opener. In the CE part, the learner wrote, "The first online learning experience in life. It was a little complicated, but I can pass for the first week." In the RO part, the learner wrote, "I can study without sleeping. They are the most understandable to me." The learner also wrote that "I have to work hard to get through this online education in the AC part." In the AE part, they wrote, "Make use of my prior knowledge so that I can fit into society in daily life." The mentor realized that this learner may have still not realized any of the EL concepts; it could also be observed in the visualization that the learner did not seem to spend enough time thinking and did not use a sentence opener. Therefore, the mentor tried to promote and support the learner to move on to learning stage by suggesting the use of sentence openers and recursive lists.

The mentor's comments and feedback were shown on the learner's user interface, as is shown in Figure 3.

Subsequently, the mentor observed that the learner may have begun to realize the concept of recursive thinking. The learner started to develop recursive learning behavior by selecting the previous experience as "None" from weeks two to four. The mentor used this information to encourage connecting to past learning and extending it into the future.

In week three, the learner started their sentence opener learning behavior (SO) through trial and error, selecting a sentence opener (*italicized text*) from the many options from the CE to the RO parts. The sentence opener from the CE part from the reflective writing data was as follows: "*Based on the experience that I had planned last week about . . .* working together with others smoothly." The sentence opener used in the RO part from the reflective writing data was as follows: "*Based on the above successful experiences, the reason that was achieved because . . .* I have found good friends to work with. We get to know each other, so my work progresses fast." The data shows that the learner started developing CE and RO thinking that week. The mentor observed that the learner started to realize the CE and RO concepts by observing their use of the CE and RO sentence openers and that they spent more time thinking about the visualization. The mentor detected a change in the thinking behavior, from a shallow to a more profound reflection on their experience, and could use such information to inform the learner and encourage them to continue doing so.

The learner started to connect the experiences in week five, as they used the experience from week four. Then, from weeks six to eight, the learner connected the experiences from the previous weeks and used the sentence opener in the CE part to describe the connection of the experience.

In week six, the learner connected the past plan of week five, "*I intend to* return to review, read various materials, and practice coding in Java programs to take the exam and collect points on Tuesday to obtain as many points as possible." This was followed by the CE part, "*Based on the experience that I had planned last week of* coding practice for test-taking, collecting points on Tuesday to get a lot of scores." The mentor recognized that the learner was aware of recursive thinking, which they learned by continuously using the recursive-thinking reminder function.

In week seven, the mentor observed that the learner had stopped using the sentence opener that described the connection from past experience but could connect it to the present. The AE in week eight showed, "*I have a plan to* diligently and persistently practice coding and learn Java in OOP to have further experience of working in coding to write faster and more accurately." The CE in week nine was, "*Based on the experience about* the diligence and determination to practice making code." The mentor observed that the learner had developed recursive learning behavior without using the sentence-opener function. Ultimately, the learner did not use the sentence opener that mentioned past experience, but recursive learning behavior could be observed from the CE, "*Based on my previous experience with* creating GUI classes, the project turned out well." In the RO, the learner wrote, "*From the successful experience mentioned above, the reason for this success is because* I and my friends in the group have returned to studying various algorithms, especially how to create a graphical user interface." The mentor could interpret that the learner may have become aware of the EL process and could therefore motivate the learner to continue this learning behavior to develop autonomous learning of EL in the future.

In the final week (week ten), in the CE part, the learner wrote, "*From the experience, I have had over the past week of* completing a project." In the RO part, the learner wrote, "*Based on that experience, the plan has accomplished* being able to write project code well and consistently, but we are still partially stuck because our group did not thoroughly study the 'inheritance' concept. We have to go back and revise it." In the AC part, the learner wrote, "Making GUI classes is not as difficult as expected. But you might feel dazed and confused at first. However, if you return to study diligently, it will be simple to comprehend." In the AE part, the learner wrote, "*I intend to* return to completing a group project and practicing code to pass the final exam with high marks".

5.1.2. Discussion on the Learner in Case Study #1

The mentor used all the available data as a communication method to show the learner's strength in understanding the EL concepts well. Based on the mentor's interpretation, the learner developed an awareness of EL as the visualization showed that they gradually progressed well by learning the EL concepts using the sentence openers. The mentor realized the quality of learner #1 could be improved, suggesting reflective writing with sentence openers. A change in thinking behavior could be detected in week three, whereas the first two weeks showed that there was no use of sentence openers at all. In week three and week four, the learner spent more time than in the first two weeks on reflective writing, using sentence openers which may indicate that the learner was thinking deeper to reflect. The mentor encouraged the learner to continue to use sentence openers for reflective writing. This behavior continued until week seven, when the learner tried to connect what they had learned previously to a new context without using a sentence opener. During the last three weeks, the learner spent less time on reflective writing using sentence openers when compared to weeks three and four. The learner also integrated the EL concepts well, and the quality of their reflective writing was excellent. Further, the learner had good self-monitoring skills, as was shown by the consistency of the MSS score (MSS-PRE: 2.90, MSS-POST: 3.97), reflective writing data, and visualization data of the learner, which shows that the learner spent time in all activities. The mentor used these data as media to communicate with the learner, demonstrating their strength and encouraging them to continue these behaviors.

Table 4 shows that the learner may have gained the confidence to use AE concepts and apply metacognitive thinking through the learning environment, as can be observed by the consistency between the reflective writing data, visualization data, and the MSS scores (MSS-PRE and MSS-POST). By using the reflection sheet, the learner may have realized the change in their attitude to be more enthusiastic about studying harder, being more diligent, and reviewing lessons more. The learner understood the EL concept more, especially in relation to AE, as they stated in the mini-exam: "Reviewing what you have learned in the past, bring the experience back, and apply it in the future".

*5.2. Case Study #2: The Learner That Became Partially Aware of EL Processes (CE and RO)*

In case study #2, we demonstrated that a learner with CE and RO as their dominant thinking behaviors became partially aware of the EL concepts (CE and RO). As presented in Figure 9, the learner spent most of their time in the CE and RO activities.

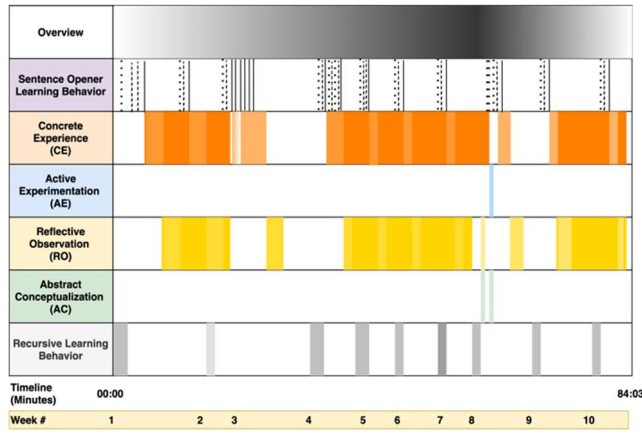

**Figure 9.** The visualization platform represents the learner's thinking behavior in case study #2.

**Table 4.** Case study #1.

| Instrument | Variable | Value |
|---|---|---|
| MSS-PRE | Mean | 2.90 |
| MSS-POST | Mean | 3.97 |
| Open-ended questionnaire (pre) | EL definition | Learning and working at the same time to gain more experience. |
| | Self-regulated learning strategies in distance learning | I will pay attention to the learning, try not to be stressed, prepare myself to review the coding after I finish each week, check the knowledge, and pay attention to get an A grade. |
| | Attitude toward distance learning | No matter in which situation, I can learn the knowledge. The attention to study is less than in the classroom, so that the understanding is lower. |
| Mini-exam | EL definition | Reviewing what you have learned in the past, bring the experience back, and apply it in the future |
| Reflection Sheet | Attitude | Before: I find it is boring and do not want to study.<br>After: Be more enthusiastic about studying. |
| | Behavior | Before: Lack of purpose and not diligence in studying<br>After: Study harder, be diligent, and review my lessons more. |
| | EL understanding | Before: I think it just a theory that learns to create the experience.<br>After: Thinking and analyzing yourself from your understanding of how you are? What are you doing today? Then understand it or not, how much, why, and what have you learned? What will be used? |
| | EL's ability | Before: I don't know anything about how to use it.<br>After: I can apply what I have learned back and apply it in the future. |
| | Reflection on … | CE: Good (4), RO: Good (4), AC: Good (4), AE: Good (4), Recursive thinking: Good (4) |
| Open-ended questionnaire (post) | Satisfaction | This learning environment changes me:<br>1. I have set more goals for studying.<br>2. I have learned that mistakes are what we need to improve ourselves.<br>3. I have been using my ideas more creatively. |

### 5.2.1. Interpretation of the Learner in Case Study #2

This learner only learned the CE and RO concepts by representing their thinking through the sentence opener that connected CE to RO. The learner started the sentence-opener learning behavior (SO) of CE and RO and began recursive learning behavior from the first week.

The mentor observed that the learner wrote their reflective writing using the sentence-opener function to connect CE to RO in week one. In the CE part, the learner wrote, "From experience with passing parameters and encapsulation. It makes me understand how to use an access modifier better." In the RO part, the learner wrote, "From that experience, what has been achieved is to choose an access modifier that is not just public. I noticed that choosing, but the public is very risky to use with everything in the project." In the AC part, the learner wrote, "The limitations of each type of access modifier and take into account the choice of use for data security. Getting the data and setting data through methods." In the AE part, the learner wrote, "More knowledge about static; how is it different from running normal methods?" A possibly educationally sound action of the mentor was to admire that the learner learned the CE and RO well, but the mentor should have also encouraged them to think about the AC and AE concepts using sentence openers. The learner continued to use the CE to RO sentence openers to represent their thinking.

In the final week (week ten), the learner wrote in the CE, "From my experience with multithread programming." In the RO part, the learner wrote, "From that experience, what was accomplished was I know that almost every program uses multithread because each program has many functions." In the AC part, the learner wrote, "Multithreading is the operation of multiple threads in a program. Threads can be sequentially defined by defining methods as synchronized. It does not work concurrently for that method." In the AE part, the learner wrote, "Practice designing and configuring threads to be the most efficient and applying them to my programs." The mentor realized that the AC part showed what the learner had learned on the subject in the AC description. The mentor might encourage the learner to think about abstracting from what they have learned from the experience rather than the concrete subject itself.

### 5.2.2. Discussion on the Learner in Case Study #2

Based on the mentor's interpretation, the learner may have only partially developed an awareness of CE and RO as the visualization showed that the learner spent most of their time in CE and RO. The learner did not integrate the EL concepts well. The quality of their reflective writing was good only in the CE and RO. The learner had poor self-monitoring, as the mentor observed the learner through their inconsistencies in visualization, reflective writing, and the MSS scores (MSS-PRE = 3.97, MSS-POST = 4.50). Further, there were inconsistencies in the reflective writing data, visualization data, and MSS scores after completing the class because the learner only learned the CE and RO. The mentor communicated with the learner to show their weak points, in which, even though the learner thought that they had good self-monitoring, observed by MSS scores, they still lacked AC skills and did not think to create the abstract concept from the experience. The learner also lacked AE thinking and did not learn how to formulate a good plan and strategy for what to learn in the future. The mentor guided the learner to spend more time on AC and AE by using the sentence openers for reflective writing and trying to learn throughout the EL cycle in the future.

### 5.3. Discussion of the Phenomena Observed in the Learning Environment

These two case studies show how the learning environment's functions can provide the data for mentors to offer learners a meaningful interpretation of their learning status. Further, the four more case studies (#3 to # 6) show the different learning processes, various learners' interpretations from the visualization, and their relationships with the MSS and reflective writing data (details omitted), as are shown in Figure 10. The visualization data was used to show how the learners grasped the EL concepts (process). Contrastingly,

reflective writing was viewed as the result (output). The MSS scores were used as an educational tool. Most mentors tended to evaluate learners based on the result [43], but the visualization showed their thinking behavior. Therefore, mentors should also monitor progress based on the EL process because it is difficult to judge based on the text alone.

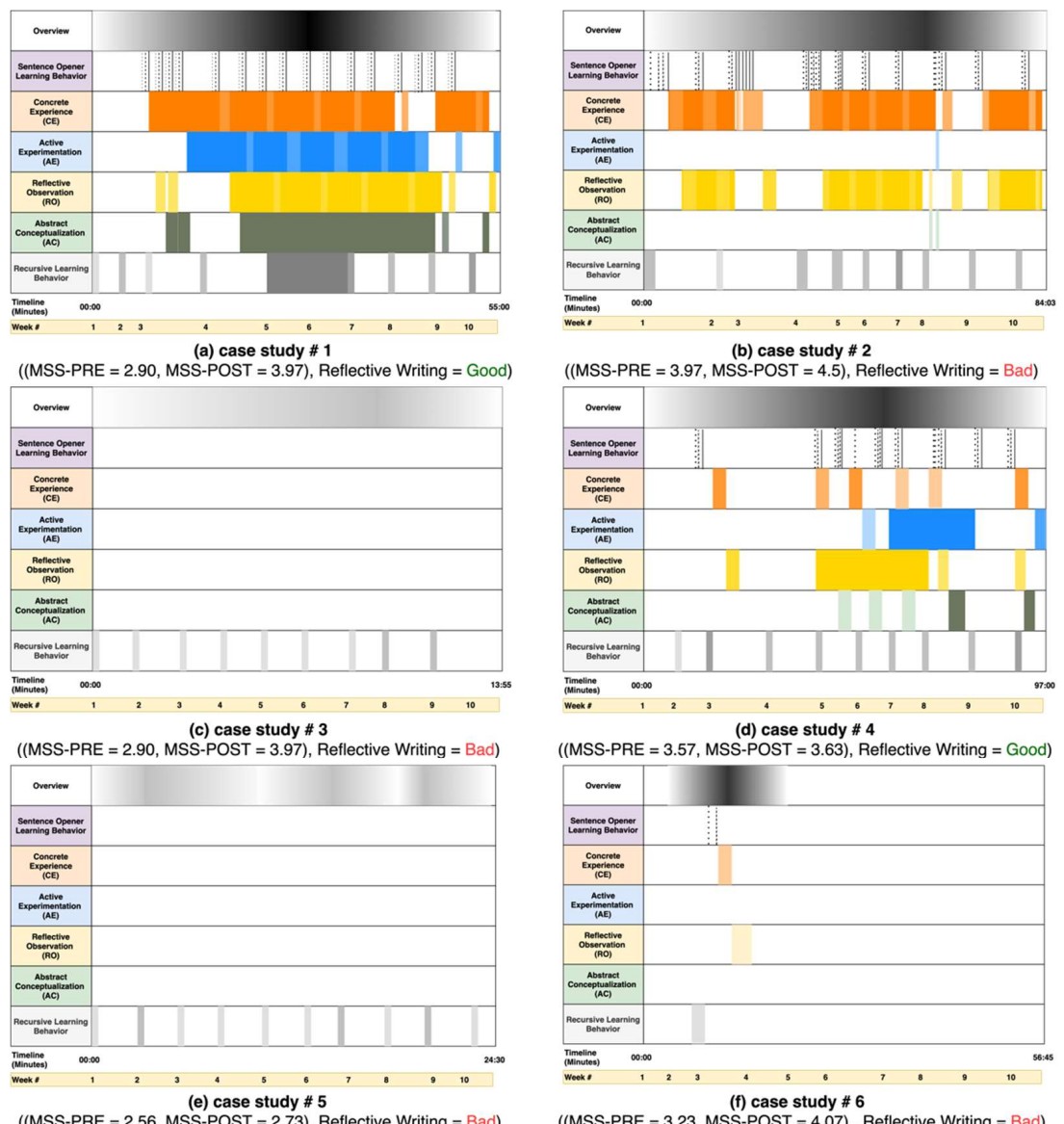

**Figure 10.** The visualization shows the two learners with other four learners and their thinking behaviors.

We demonstrate how the mentor used visualization data as observation interpretation and communication tool with the learner. If the mentor used the MSS data only, then case study #1 (MSS-PRE = 2.90, MSS-POST = 3.97, Reflective writing = Good) and case study #3 (MSS-PRE = 2.97, MSS-POST = 4.03, Reflective writing = Bad) are the same (i.e., the mentor cannot differentiate them). Case study #3 may demonstrate the Dunning–Kruger effect [44], in which incompetent learners tend to overestimate their skills and abilities, because the visualization data shows that the learner only used the recursive reminder function with poor reflective-writing quality. In case study #5 (MSS-PRE = 2.56, MSS-POST = 2.73, Reflective writing = Bad), the mentor could praise learners who were well self-monitored, even when their reflective writing was not good, but the learner may yet realize the poorness of their performance. Case studies #2 (discussed in section B), #4, and #6 show learners who were not well-self-monitored. For

the learner in case study #4 (MSS-PRE = 3.57, MSS-POST = 3.63, Reflective writing = Good), results show that this learner initially overestimated their metacognitive thinking, so the mentor showed the inadequate self-monitoring during the pre-learning phase. Case study #6 (MSS-PRE = 3.23, MSS-POST = 4.07, Reflective writing = Bad) overestimated their metacognitive thinking, so the mentor commented on their inadequate self-monitoring by reflecting on their EL using sentence openers and encouraged their improvement in the future.

## 6. Conclusions

This study proposed a computerized learning environment to raise learners' awareness of the EL process by promoting their metacognitive thinking through reflective writing support. In this learning environment, learners connected their real-world experiences through the EL process by using reflective writing to gain a new understanding. Our online reflective writing framework gives learners the opportunity to follow EL processes by answering the EL-guided questions to realize the input and output of each process. The recursive-thinking reminder function reminds learners to connect what they have experienced in the past to the present and then apply what they have learned to the future. The EL process verbalization support function using sentence openers is designed to play a dual role. The first role is to provide a language to stimulate metacognitive thinking about the EL concepts, which allows learners to reflect on their experience using sentences that represent their thoughts. The second role is to provide a learning sensor that senses actions and transforms those actions into thinking behaviors by externalizing the implicit thinking behavior, giving mentors additional information that can be observed by visualizing learners' thinking-behavior functions. We demonstrate how learners learn through the learning environment and how mentors can use the information available to support those learners. The view that writing can promote thinking and help monitor the learning process can be generalized to a metacognitive-thinking support framework applied in other problem domains, such as internship activities, active classroom learning, distance learning, project-based learning, and new employee training. Consequently, society can become a knowledge-based society that applies metacognitive thinking.

In this study, a short period (a ten-week course) was used for learning through the learning environment. The learners' acquisition of metacognitive skills cannot be accomplished in such a short time; these skills must be acquired through daily activities, such as learning in class, internships, and jobs. With this limitation, we need to acquire data by operating this framework for an extended time scale to validate the sentence openers and evaluate the learning outcomes. To validate sentence openers on the quality of language as teaching material and evaluate the learning outcomes, we currently operate this framework in many situations in university education (such as an internships and distance learning) and desire to evaluate the learning effect in the future. Therefore, we plan to investigate this learning environment over an extended period of multiple-phased internships in Panyapiwat Institute of Management. This plan involves stakeholders, including multiple companies with different work environments in different internship phases and mentors' skills supporting experiential learning. These challenges can provide an excellent setting to conduct future research.

To fully support the educational value of this learning environment through the visualization of the learners' thinking behavior functions, future research could further investigate whether open-learner models [45] could represent the learner as a necessary means for learning support. Open-learner models aim to provide learners with learning-process data that enables them to view information about themselves. Such models may help us to conceptualize which types of learners' thinking behaviors can be used as criteria to classify learners. The learners can view and reflect on the visualization of their learning process by themselves, as sometimes it may not reflect what they are; this promotes self-reflection and self-monitoring. This model can be used as a tool to communicate with

mentors to co-create values. In this study, the visualization data was only visible to the mentor. Therefore, we should evaluate the visualization before providing it to learners.

The learning environment is an online system that stores data in a database. These data can be used for further analysis; for example, how learners use each sentence opener. For sentence-opener refinement, we can use that data to decide whether they should be removed, revised, or added. We could also track whether new sentences developed from the learners' reflective writing could be used as options in future experiments. This future work would ensure the quality and validity of each sentence and provide co-creation opportunities between learners and mentors.

Finally, teaching the mentors how to support learners' thinking about EL is another good future research direction to assess how mentors provide feedback on the learners' self-monitoring of their metacognitive thinking. Mentors can use the reflective writing description, visualization platform, and learners' self-monitoring to provide the learners with suitable feedback.

**Author Contributions:** Conceptualization, M.I. and K.T.; methodology, C.K., M.I., K.T., K.M., T.T. and T.S.; software, C.K. and P.C.; investigation, C.K. and P.C.; resources, T.S. and P.C.; data curation, C.K.; writing—original draft preparation, C.K., K.M. and M.I.; writing—review and editing, T.T., K.T., P.C. and T.S.; visualization, C.K., K.M. and M.I.; supervision, M.I. and T.T.; project administration, T.S., T.T., M.I. and K.T.; funding acquisition, T.S., T.T., M.I. and K.T. All authors have read and agreed to the published version of the manuscript.

**Funding:** This research was funded by JSPS KAKENHI under grant number JP18H01050, the Thailand Research Fund under grant number RTA6080013, the TRF Research Team Promotion Grant (RTA), and the Thailand Research Fund under grant number RTA6280015.

**Institutional Review Board Statement:** Not applicable.

**Informed Consent Statement:** Informed consent was obtained from all subjects involved in the study.

**Data Availability Statement:** Data sharing not applicable.

**Acknowledgments:** The authors would like to thank the reviewers for their thoughtful comments and efforts towards improving this manuscript; the Panyapiwat Institute of Management for supporting this study; the Research Fund of Japan Advanced Institute of Science and Technology (JAIST), Japan; the Research Fund of the Sirindhorn International Institute of Technology (SIIT) Thammasat University; the Research Fund of Center of Excellence in Intelligent Informatics, Speech and Language Technology and Service Innovation (CILS); the Intelligent Informatics and Service Innovation (IISI) Research Center; Thailand's National Electronics and Computer Technology Center (NECTEC); and the National Science and Technology Development Agency (NSTDA).

**Conflicts of Interest:** The authors declare no conflict of interest.

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
