# Peer review of "A Learning Environment to Promote Awareness of the Experiential Learning Processes with Reflective Writing Support"

_education, doi:10.3390/educsci13010064_

Round 1
Reviewer 1 Report
It presents in the introduction, lines 44-57, specific aspects of the study carried out and appropriate in other sections within this study such as in data analysis or in the discussion/conclusions. You should present the results of other studies that would support the conduct of this research in the introduction.
Point 3.1.2., lines 206-233, is appropriate for the methodology, and table 1, for the analysis of the data.
The article presents a detailed description of the research process and the experience carried out that should be highlighted, but does not indicate this as a point of methodology. It would be necessary to include a section where all the methodological aspects fit in, such as various parts of point 3 and point 4. It is recommended to review lines 178-281 in section 3 which would be part of the methodology.
The Likert scale indicated in line 278 should be changed to a graphic or conceptual scale, because the Likert questionnaire is a questionnaire that seeks statistical selection of the process that is asked in its items, and this is not done by the authors in his study.
It is recommended to order the different sections to clearly and explicitly indicate the methodology carried out.
Reviewer 2 Report
Overall, an interesting paper that presents a potentially very useful learning platform for supporting experiential learning and reflective writing in educational programmes.
While the background and contextual information seems to be quite well researched and articulated, the authors do not appear to have considered any other models of reflective learning, instead settling on Kolb’s EL model as the most well-known. Could they indicate whether any other models were examined, and if so, why they were not chosen as the framework for this study? Examples might be Gibbs’ Reflective Learning Cycle, or Rolfe, Freshwater & Jasper’s framework. Was Kolb’s model evaluated in terns of its user-friendliness for students, who are unfamiliar with the idea of learning from experience, and expressive this through reflective writing? It would be interesting to see a more robust justification for choosing this specific model, as the concepts are complex and potentially difficult to grasp (although it is noted that an orientation phase is built-in to familiarize learners with the cycle, and guided questions and prompts are used in the learning environment to help with this).
Does Table 1 simply offer the authors’ estimations of how learners might reflect, or is it based on learners’ actual comments?
The idea of the supportive online learning environment is interesting, and the fundamental structure of the platform/interface seems solid, especially the available support functions (sentence openers, etc.) However, I found the descriptive account of the learning environment design difficult to follow, especially section 4.1. I feel that the description of the process here could be simplified and made more accessible to the general reader. The extensive use of initials in this section adds to the lack of clarity, and it’s easy to become lost in the text here. The idea of “dimensions” is also introduced here without any preamble or context. However, by contrast, the interface depicted in Figure 3 was extremely helpful in illustrating how it appears to learners, and works in practice, and this brought some clarity to the complex description.
The visualization function for mentors (which seems to more or less fall into the category of learning analytics) is an interesting element of the learning environment, and an undoubtedly useful feature of the platform – however again here, the descriptive account is dense and difficult to follow, although helped somewhat by the visuals. I feel that this would be challenging for the general reader. The two case studies are more helpful, in that they allow the readers to connect the learners’ experiences to the visual representation generated by the data analytics. However, aspects of these are also difficult to follow, and would potentially lose readers.
I also noted some unusual phrasing in places, which further proof-reading will catch – for example, “imbibing” experiential learning.
In general, my feeling was that, while interesting, the paper is overly long and complex, with dense description that obscures the key important messages of the study. Perhaps it was the case that the authors were endeavoring to write the paper to appeal to both instructors and instructional designers, whereas a different approach is needed for each audience. I think there is much work to be done here to improve the readability and accessibility of the paper, and therefore to increase its potential impact.
Round 2
Reviewer 2 Report
The authors have done a very good job in responding to the comments in the original review (reviewer 2), and the resulting paper is more coherent, easy to read and flows well. Overall, while there are some parts that may be challenging to follow by readers who are unfamiliar with the data visualization and analytical methods used, the general methodological description and presentation of results are better articulated and more accessible to the average reader. The paper presents a useful account of an interesting learner interface that promotes experiential learning through the effective use of prompts and mentorship to support reflective writing.